# COVID-19: Post-recovery long-term symptoms among patients in Saudi Arabia

**Mostafa M. Khodeir**[1,2]*, **Hassan A. Shabana**[3], **Zafar Rasheed**[4]*, **Abdullah S. Alkhamiss**[2], **Mohamed Khodeir**[5], **Mohammad S. Alkhowailed**[6], **Sami Alharbi**[7], **Mansour Alsoghair**[8], **Suliman A. Alsagaby**[9], **Waleed Al Abdulmonem**[2]

**1** Faculty of Medicine, Department of Pathology, Cairo University, Cairo, Egypt, **2** Department of Pathology, College of Medicine, Qassim University, Buraidah, Qassim, KSA, **3** Faculty of Medicine, Chest Diseases Department, Cairo University, Cairo, Egypt, **4** Department of Medical Biochemistry, College of Medicine, Qassim University, Buraidah, Qassim, KSA, **5** Psychiatric Department, Brook Hospital, Louisville, Kentucky, United States of America, **6** Department of Dermatology, College of Medicine, Qassim University, Buraydah, Qassim, Saudi Arabia, **7** Pulmonary Department, King Fahad Specialist Hospital, Ministry of Health, Buraidah, Saudi Arabia, **8** Department of Family and Community Medicine, College of Medicine, Qassim University, Buraidah, Qassim, KSA, **9** Department of Medical Laboratories, Central Biosciences Research Laboratories, College of Science in Al Zulfi, Majmaah University, Al Majma'ah, Saudi Arabia

* dr.mustafa31@gmail.com (MMK); zafarrasheed@qumed.edu.sa, zrasheed@qu.edu.sa (ZR)

**Data Availability Statement:** All relevant data are within the manuscript.

**Funding:** The authors received no specific funding for this work.

## Abstract

### Background

After recovery from acute infection with severe acute respiratory syndrome coronavirus 2 (SARS-CoV-2), many patients experience long-term symptoms in different body systems. The aim of the present study was to identify these symptoms, their severity, and their duration as a first step in building a system to classify post-recovery long-term symptoms of coronavirus disease 2019 (COVID-19).

### Methods

An online-based cross-sectional survey was administered between September and October 2020. Data regarding the severity of post-recovery symptoms and their duration were collected using an Arabic questionnaire divided into six categories encompassing the 20 most prevalent symptoms.

### Results

A total of 979 patients recovered from COVID-19 in Saudi Arabia in the study period, of whom 53% were male and 47% were female. The most common symptoms included general fatigue and weakness (73% each), with moderate severity of neurological symptoms including mood changes (41%) and insomnia (39%). Among the special senses, loss of smell and taste of marked severity were reported by 64% and 55% among respiratory symptoms, cough of mild severity (47%), and dyspnea of moderate severity (43%). Loss of appetite of moderate severity was reported in 42%, and diarrhea, abdominal pain, and nausea of mild severity were reported by 53%, 50%, and 44% of respondents, respectively.

**Competing interests:** No authors have competing interests.

## Conclusions

Long-term symptoms after recovery from COVID-19 warrant patient follow-up. The authors propose a classification system as a starting point to guide the identification and follow-up of long-term symptoms post-recovery, and recommend larger-scale studies to broaden the definition of recovery from COVID-19, which appears to have two phases, acute and chronic.

## Introduction

An outbreak of a novel coronavirus, putatively termed severe acute respiratory syndrome coronavirus 2 (SARS-CoV-2), the causative agent of coronavirus disease 2019 (COVID-19), occurred in the city of Wuhan, China in December 2019 [1]. In March 2020, the World Health Organization declared the outbreak a pandemic. The identified virus, SARS-CoV-2, is similar to other fatal coronaviruses, namely SARS-CoV and Middle East Respiratory Syndrome coronavirus [2,3].

Coronavirus infection commonly causes respiratory and gastrointestinal symptoms; similarly, SARS-CoV-2 infection also leads to such symptoms, mainly fever, generalized weakness, lower respiratory tract symptoms, and anosmia caused by severe lung injury and acute respiratory distress syndrome, and affects multiple organs possibly causing organ failure, especially in old patients (>60 years) patients and/or those with specific comorbidities, which suggests that COVID-19 should be considered a systemic disease [4–6].

Many studies and reviews have focused on systems affected other than the respiratory system, including neurological, cardiac, vascular, gastrointestinal, kidney, and cutaneous manifestations [7–17]. Aside from the acute clinical challenges in those with severe illness, the long-term sequelae of pneumonia have been reported in many studies focusing on the decline in quality of life after pneumonia diagnosis that did not return to baseline—even after 1 year—and persistent late psychological effects, alongside the implications for rehabilitation and health care utilization [18–21]. The long-term effects of viral pneumonia among survivors, including those infected with SARS and MERS, have been documented. The same has been observed in those who recovered from COVID-19, in which survivors of acute illness experience long-term symptoms that persist for varying periods and varying degrees of severity. However, there are only a limited number of studies that followed patient symptoms after recovery from acute illness with COVID-19, and most focused on only one or a few symptoms, such as anosmia and cardiac conditions [22–27]. These symptoms impose a burden on patient health [28], and also affect physical and social status [29,30], which in turn have adverse effects on virtually all aspects of human activity and contribute to high burden on the economy and society. The pathogenesis of these post-recovery, long-term symptoms of COVID-19 remains unclear. As such, there is an urgent need to categorize these long-term symptoms according to the severity of their impact. This will raise awareness of the urgent need for studies to determine the underlying pathogenesis of the disease to better understand the process and, secondarily, for proper planning for patient management.

To date, published studies have focused on only one or a few symptoms, and have been smaller-scale investigations, which may increase the risk for missing other symptoms that lead to more serious longer-term effects on patients. Accordingly, the aim of the present study was to survey a larger number of patients who recovered from acute infection and illness with SARS-CoV-2/COVID-19 using different variables and an adequate number of post-recovery

symptoms as a first step in building a classification system for COVID-19 post-recovery symptoms according to significance of importance, which should provide an opportunity for deeper understanding and future research to fill important knowledge gaps.

## Materials and methods

### Study design

Ethical approval for this study was obtained from the Subcommittee of Health Research Ethics, Deanship of Scientific Research, Qassim University (ref. # 20-03-04) and informed consent was obtained from all participants during the online-based cross-sectional survey. A survey was administered between September and October 2020 using Google Forms and Twitter as a forum, an anonymous Arabic survey was distributed. Saudi nationals and residents who had been recorded to be recovered from acute COVID-19 symptoms have been included in the study. There were no monetary benefits and participation was purely voluntary. Participants were told about the purpose of the questionnaire, the sample, and the significance of the score for each answer on the survey's landing page, and were required to provide informed consent before proceeding to respond to the items. Participants were thanked for their participation on the final page. A new Twitter account was used to recruit the participants. Individuals and organizations received "tweets" requesting that they "retweet" the survey connection. According to Internet protocol restrictions, multiple enrolments by the same individual were prohibited.

### Instrument development

The questionnaire used in this study was created using a combination of published literatures [31–35] and internal discussion among the research team to assess question format, comprehensiveness, clarity, and flow. Participants were assured that their responses would be collected anonymously, reducing the potential for bias introduced by self-reported results. The questionnaire was created and required approximately 3–5 minutes to complete with the aim of reducing survey fatigue. It was tested for validity using face validation by the survey research experts. The questionnaire was written in English by a native English speaker and translated into Arabic by two native Arabic speakers. The Arabic version of the questionnaire was piloted and distributed.

### Instrument measures

The model was conducted on a group of 40 recovered COVID-19 cases in order to optimize the wording and clarification of the survey questions. The survey was widely distributed after slight changes to the format and vocabulary. The results of the pilot study were not included in any subsequent research but used for construct validity by employing the Pearson's correlation coefficient $r$ of the scores of respondents' responses to an item with their total scores. According to the classification of r values as previously reported [36], the results showed that none of the questions presented poor correlations. All questions showed positive correlation that ranged from fair to very strong with significant relationships (Pearson's r = 0.34–1.00; p<0.05). The instrument's reliability was verified through Cronbach's alpha as described in previously [37] by calculating the existing correlations. The result of this test was 0.870, which reflected a strong internal consistency. Final questionnaire was structured into 6 sections addressing specific symptoms potentially correlated with SARS-CoV-2 infection/COVID-19 as follows: (I) General symptoms (2 items: fatigue and weakness); (II) Skin and musculoskeletal symptoms (3 items: muscle ache[s], joint pain, and skin rash); (III) Psychological and neurological

symptoms (4 items: headache, mood changes, insomnia, esthesia, and anesthesia); (IV) Special sense symptoms (5 items: hearing problems, visual disturbances, dry eyes, loss of smell, and loss of taste); (V) Respiratory system symptoms (3 items: cough, shortness of breath, and chest tightness); and (VI) Gastrointestinal symptoms (4 items: lack of appetite, nausea, diarrhea, and abdominal pain). Each section consisted of a group of related symptoms. For each symptom, participants were asked to score the severity of each on a three-point scale (mild, moderate, and severe, scored as 1, 2, and 3, respectively), and to report the duration of persistence of this symptom(s) after recovery in days. More than one symptom could be reported. Information regarding participant demographics and the method of diagnosis of SARS-CoV-2 infection was collected.

## Sample size calculations

The estimated sample size was 384, which was obtained by a statistical calculation from the official Saudi Ministry of Health records, announced the number of recovered cases with a 95% confidence level and a 5% margin of error. We applied the online survey from September to October 2020 and a total of 992 Saudi subjects were approached. Out of them, 979 were included and the rest 13 subjects were omitted as they were made invalid selection. As reported previously [38,39], that sample size of 500 is very good, and 1000 or more is excellent, larger samples are always better than smaller, therefore it is recommended to utilize as large a sample size as is possible. Applying the same principle, 979 Saudi nationals were recruited in the study.

## Data management and statistical analysis

The datasets were processed and analyzed using the PivotTable, data analysis within Microsoft excel version 2019. Descriptive statistics such as frequencies, percentage, mean and standard deviation (SD) were employed for the presentation of categorical and continuous variables to summarize respondent characteristics. Moreover, the data were further verified by SPSS version 25.0 (IBM Corporation, Armonk, NY, USA) using ANOVA and LSD tests. Differences with $p \leq 0.05$ were considered to be statistically significant. Bar graphs were plotted for different variable including the age groups, methods of diagnosis, gender proportion and average days for each of the COVID-19 post recovery long-term symptoms.

## Results

Of the 979 patients recovered from COVID-19, as they were post recovery group so none were experiencing acute symptoms or fever at the time the survey was administered. The males and females respondents in the study were 53% and 47%, respectively. Characteristics of the study population are summarized in Table 1. The frequency, degree of severity, and persistence of symptoms (in days) after recovery are summarized in Tables 2 and 3, and Figs 1–3. The largest age group comprised individuals 20 to 39 years of age (57.1%), with a mean age of 37.69. The most common long-term (i.e., persistent) symptoms were of general symptoms group, fatigue and weakness (73% each), which persisted for a mean of 7 and 8.11 days respectively, while muscle aches were the dominant symptom (66%) among the skin and musculoskeletal symptom groups, with a mean persistence of 7.8 days. Among psychological and neurological symptoms, headache was the most common (64%), with a mean persistence of 6.5 days. The presenting symptom among the special sense group was loss of smell (62%), with a mean persistence of 15.9 days (Table 2). Cough was the dominant symptom (47%) in the respiratory system group, and lack of appetite (46%) was the dominant symptom in the gastrointestinal group, with a mean persistence of 11 days and 9.4 days, respectively (Table 2). The most severe

**Table 1. Demographic and clinical characteristics of the study individuals.**

| Characteristics | Value |
|---|---|
| Age, years (y)—mean (±SD) | 37.69 (±10.77) |
| Minimum | 10 y |
| Maximum | 84 y |
| Gender | Number of subjects (N) (%) |
| Male | 519 (53%) |
| Female | 460 (47%) |
| Method of diagnosis | N (%) |
| Swab only | 117 (12%) |
| Symptoms only | 158 (16%) |
| Swab & Symptoms | 704 (72%) |
| Age groups by years | N (%) |
| 10–19 | 31 (3.2%) |
| 20–29 | 177(18.1) |
| 30–39 | 383(39.1%) |
| 40–49 | 241(24.6%) |
| 50–59 | 113(11.5%) |
| 60–69 | 30 (3.1%) |
| 70–79 | 3 (0.3%) |
| 80–89 | 1(0.1%) |
| Age group categories | N (%) |
| >20 y | 32 (3.3%) |
| 20–39 y | 559 (57.1%) |
| 40–59 y | 354 (36.2) |
| 60 and above | 34 (3.5%) |

degree of symptom was scored "3", which was recorded for both loss of smell and loss of taste, which also demonstrated the highest mean duration of symptom persistence (15.9 and 13.9 days, respectively) (Table 3). To study these findings in depth, we further categorized the studied cases in four age groups (Table 1). We observed a significant correlation of post-recovery COVID-19 symptoms with age, persistence of symptoms and degree of severity. Furthermore, we also found a statistically significant relation of age with the presence of post-recovery COVID-19 long term symptoms degree of severity and/or persistence, such as weakness degree (p = 0.003), persistence (p = 0.001), lack of appetite degree (p = 0.02), persistence (p = 0.003), Insomnia degree (p = 0.01), loss of smell degree (p = 0.002), loss of taste degree and Headache degree (p = 0.04 each), cough degree (p = 0.01) significant correlation was found between age and persistence of symptoms as fatigue (p = 0.004), joint pains (p = 0.01), mood changes (p = 0.03), nausea (p = 0.002) and abdominal pain (p = 0.02; Tables 2 & 3). According to the findings we proposed a scoring system that can be can be delivered online for post discharge follow-up, and the score of each case can be automatically calculated. The case that will reach the proposed score should be invited for follow up in the clinic, keeping in mind high concerns groups (as in Table 4 level 1A&B) should be the first priority follow up group, while lower concerns groups (as in Table 5 level 2 A&B) should be the second priority follow up. The cases will not reach the proposed score, no need for clinic follow up and should be reassured as mostly their symptoms are self-limited. By this scoring system we can easily pick up patients of concern to be followed up and also decrease load on health services as much as possible.

**Table 2. Persistence days of post-recovery of COVD-19 recorded symptoms in 979 studied subjects.**

| | Descriptive Statistics | Mean (days) | Median (days) | Standard Deviation | Range (days) | No. | % | p-value |
|---|---|---|---|---|---|---|---|---|
| **I-General Symptoms** | Fatigue Days | 7 | 5 | 8.6 | 120 | 719 | 73% | 0.004* |
| | Weakness Days | 8.1 | 5 | 9.8 | 119 | 715 | 73% | 0.001* |
| **II- Skin and Musculoskeletal symptoms** | Muscle aches Days | 7.8 | 5 | 12 | 119 | 646 | 66% | 0.14 |
| | Joint pains Days | 7.2 | 5 | 9.4 | 119 | 595 | 61% | 0.01* |
| | Skin Rash Days | 5.4 | 2 | 11.7 | 119 | 164 | 17% | 0.62 |
| **III- Psychological and Neurological symptoms** | Headache Days | 6.5 | 4 | 8.2 | 119 | 627 | 64% | 0.73 |
| | Mood changes Days | 9.5 | 7 | 11.9 | 154 | 501 | 51% | 0.03* |
| | Insomnia Days | 8.8 | 7 | 10.6 | 89 | 432 | 44% | 0.21 |
| | Paraesthesia and anesthesia Days | 7 | 4 | 11.1 | 119 | 296 | 30% | 0.85 |
| **IV- Special sense symptoms** | Hearing problems Days | 7.5 | 5 | 11.9 | 119 | 129 | 13% | 0.94 |
| | Visual disturbances Days | 7.3 | 4 | 9.9 | 59 | 145 | 15% | 0.40 |
| | Dry eyes Days | 7.9 | 5 | 11 | 63 | 185 | 19% | 0.88 |
| | Loss of smell Days | 15.9 | 10 | 19.5 | 149 | 604 | 62% | 0.63 |
| | Loss of taste Days | 13.9 | 9 | 16.6 | 119 | 555 | 57% | 0.60 |
| **V- Respiratory system symptoms** | Cough Days | 11 | 7 | 12.8 | 120 | 457 | 47% | 0.09 |
| | Breathlessness and chest tightness Days | 9 | 6 | 10.7 | 119 | 388 | 40% | 0.45 |
| **VI- Gastrointestinal symptoms** | Lack of appetite Days | 9.4 | 7 | 8 | 59 | 455 | 46% | 0.003* |
| | Nausea Days | 6.9 | 5 | 7 | 59 | 313 | 32% | 0.002* |
| | Diarrhea Days | 6.4 | 4 | 8.8 | 119 | 402 | 41% | 0.42 |
| | Abdominal pain Days | 6 | 4 | 6.6 | 44 | 253 | 26% | 0.02* |

*p<0.05 considered statistically significant. P values were calculated by ANOVA and LSD tests using SPSS software.

The mean (±SD) age of the subjects was 37.69 (±10.767) years.

## Discussion

To our knowledge, the present study is the first to describe the duration and severity of symptoms among individuals in Saudi Arabia following infection with SARS-CoV-2 and subsequent illness with COVID-19. This study was designed to address post-recovery symptoms in patients who experienced acute symptoms of COVID-19.

Twenty symptoms were included in the questionnaire and covered general symptoms, and skin, musculoskeletal, neurological and psychological symptoms. Fatigue, and weakness of moderate severity were the most commonly reported general symptoms (73% each), with a mean duration of 7 days for fatigue; however, symptoms of weakness persisted longer, with a mean duration of 8.1 days. These symptoms are usually observed in recovering patients with respiratory tract infections among old age (>60 years) patients, or after lengthy hospital stay or critical illness [40–45].

More than 80% of our participants comprised a relatively young age group (20 to 50 years) with good health status before contracting COVID-19, which indicates that weakness and fatigue are significant post-recovery symptoms of COVID-19, and is consistent with other studies that described fatigue as a long-term symptom and one of several common sequelae of COVID-19. Moreover, it is independent of the severity of the previous acute illness or levels of pro-inflammatory markers [46]. Muscle ache(s) of moderate severity and joint pain of mild severity were reported by 43% and 39% of respondents, respectively. Myalgia during the acute stages of COVID-19 has been well documented [47], ranging from 11% to 44% in other studies

**Table 3. Degree of severity of the recorded post recovery COVD-19 long-term symptoms with their frequency of expression.**

| Symptoms groups | Long term recorded symptom | Degree of severity and its frequency of expression | | | No. | p-value* |
|---|---|---|---|---|---|---|
| | | Mild N (%) | Moderate N (%) | Severe N (%) | | |
| **I-General Symptoms** | Fatigue | 239 (33%) | 326(45%) | 154(21%) | 719 | 0.05 |
| | Weakness | 254(36%) | 328(46%) | 133(19%) | 715 | 0.003* |
| **II- Skin and Musculoskeletal symptoms** | Muscle aches | 208 (32%) | 279(43%) | 159(25%) | 646 | 0.36 |
| | Joint pains | 232(39%) | 220(37%) | 143(24%) | 595 | 0.44 |
| | Skin Rash | 142(87%) | 17(10%) | 5(3%) | 164 | 0.86 |
| **III- Psychological & Neurological symptoms** | Headache | 195(31%) | 242(39%) | 190(30%) | 627 | 0.04* |
| | Mood changes | 166(33%) | 216(43%) | 119(24%) | 501 | 0.20 |
| | Insomnia | 157(36%) | 171(40%) | 105(24%) | 432 | 0.01* |
| | Paraesthesia and anesthesia | 167(56%) | 96(32%) | 33(11%) | 296 | 0.25 |
| **IV- Special sense symptoms** | Hearing problems | 95 (74%) | 23(18%) | 10(8%) | 129 | 0.41 |
| | Visual disturbances | 108(75%) | 25(17%) | 12(8%) | 145 | 0.34 |
| | Dry eyes | 108(58%) | 59(32%) | 18(10%) | 185 | 0.01* |
| | Loss of smell | 95(16%) | 124(21%) | 385(64%) | 604 | 0.002* |
| | Loss of taste | 95(17%) | 152(27%) | 308(55%) | 555 | 0.04* |
| **V- Respiratory system symptoms** | Cough | 215(47%) | 149(33%) | 93(20%) | 457 | 0.01* |
| | Breathlessness and chest tightness | 159(41%) | 166(43%) | 63(16%) | 388 | 0.42 |
| **VI- Gastrointestinal symptoms** | Lack of appetite | 138(30%) | 191(42%) | 126(28%) | 455 | 0.02* |
| | Nausea | 137(44%) | 108(34%) | 68(22%) | 313 | 0.22 |
| | Diarrhea | 212(53%) | 120(30%) | 70(17%) | 402 | 0.69 |
| | Abdominal pain | 127(50%) | 103(41%) | 23(9%) | 253 | 0.83 |

*p<0.05 considered statistically significant. P values were calculated by ANOVA and LSD tests using SPSS software.

The mean (±SD) age of the subjects was 37.69 (±10.767) years.

[48,49], and myalgia and arthralgia were also evident in various other coronavirus infections during acute illness [50].

The persistence of myalgia and arthralgia after acute illness may reflect the activation and triggering of excessive or uncontrolled cytokine responses, and a local inflammatory reaction

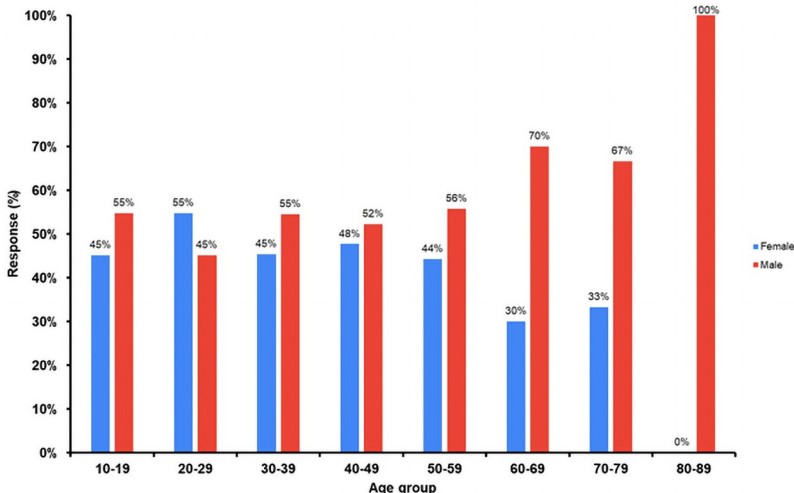

**Fig 1. Age groups and gender proportion for recovered COVID-19 patients.**

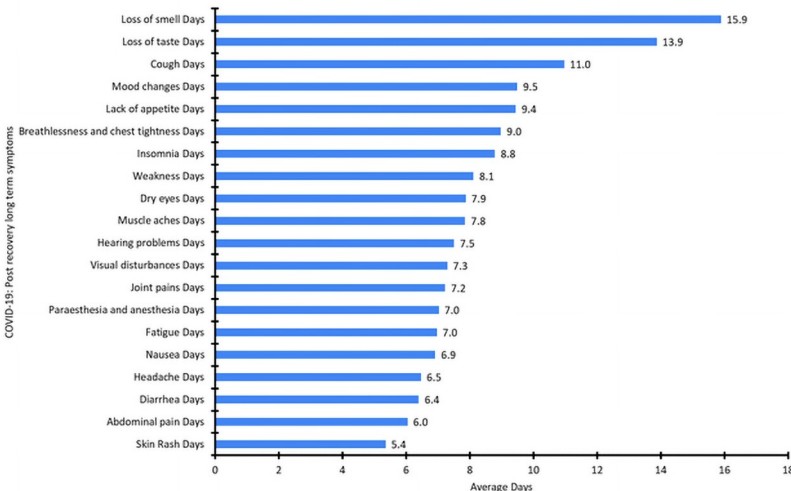

**Fig 2. Average days for different COVID-19 post recovery long term symptoms.**

in the respiratory tract, especially the alveoli, which progresses to involve other organs, especially the joints and muscles [51].

Neurological complications have also been studied in individuals infected with other human coronaviruses. Neuro-invasion causing neurological pathologies ranging from headache and anosmia to severe and fatal encephalopathy, encephalitis acute myelitis, Guillain-Barré syndrome, and other cerebrovascular pathologies have been documented [52].

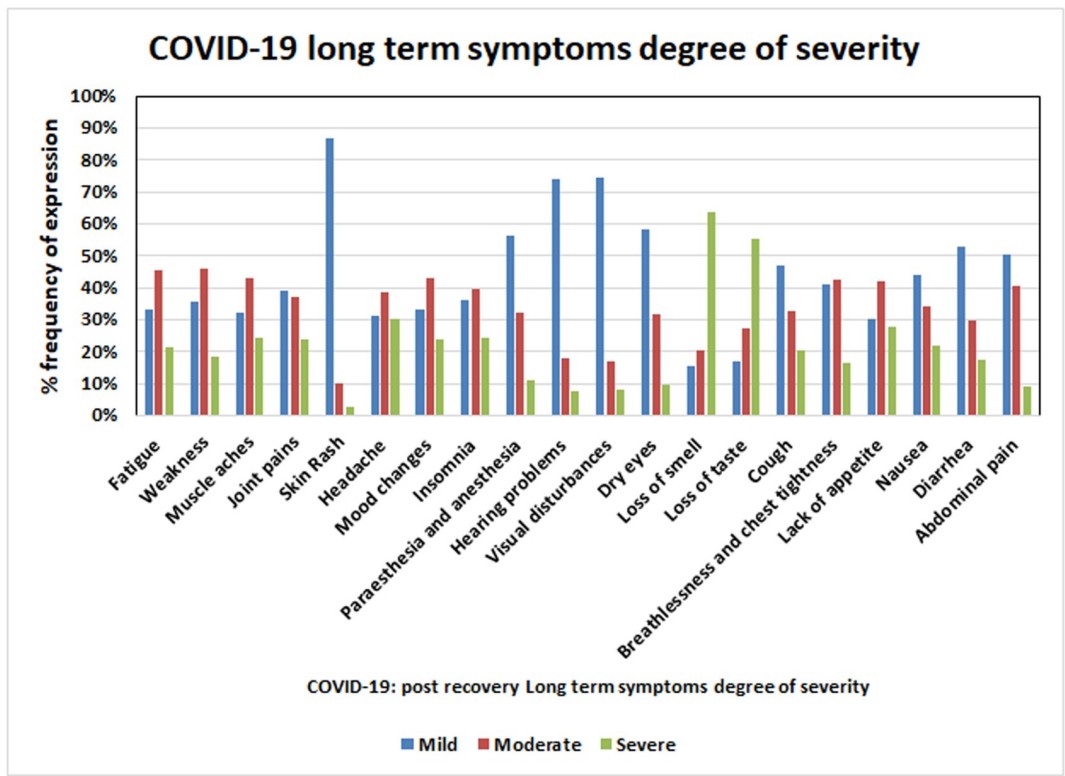

**Fig 3. Degree of severity and frequency expression for different COVID-19 post recovery long term symptoms.**

**Table 4. (Level-1 A): Scoring Post-recovery COVID-19 long term symptoms persistence (High concerns groups).** Scoring method: One point given for each item per symptom (one point for days and one point for age group). Interpretation: Score two for any symptom: follow up is recommended for the patient for each symptom got score 2. **(Level 1 B): Scoring Post-recovery COVID-19 long term symptoms degree of severity (High concerns groups).** Scoring method: One point given for each item per symptom (one point for severity and one point for age group). Interpretation: Score two for any symptom: follow up is recommended for the patient for each symptom got score 2.

| Long term symptom | Days ($\geq$)* | Age groups (years, y) |
|---|---|---|
| Fatigue | 5 | $\geq$41 |
| Weakness | 5 | $\geq$40 |
| Joint pains | 5 | $\geq$20 |
| Mood changes | 7 | 20-59y |
| Lack of appetite | 7 | 20-59y |
| Nausea | 5 | 20-59y |
| Abdominal pain | 4 | 20-59y |
| **Long term symptom** | **Degree of severity ($\geq$) ** ** | **Age group** |
| Fatigue | 2 | Any age group* |
| Weakness | 2 | Any age group* |
| Headache | 2 | 20-59y |
| Insomnia | 2 | <20y or $\geq$ 40 y |
| Dry eyes | 1 | 20–39 or $\geq$60y |
| Loss of smell | 3 | 20–39 or $\geq$40y |
| Loss of taste | 3 | 10-39y |
| Cough | 1 | <20y or 40–59 y |
| Lack of appetite | 2 | <20y or $\geq$40y |

*Days here are the recorded median number for each symptom.

*Cases age between 10–84 years.

**Degree of severity Mild = 1, Moderate = 2, Severe = 3.

In the present study the post-COVID-19 headache was reported by 64% of our participants, it was of moderate severity among 39% and persisted for a mean of 6.5 days, whereas anosmia was reported in severe degree by 64% and persisted for a mean of 15.9 days. Headache is considered one of the characteristic and cardinal symptoms of COVID-19, and is observed in 68.3% of patients in the emergency department [53]. Other symptoms addressed in our survey included mood changes of moderate degree in 43% and paresthesia in 56% of mild cases.

Anosmia is currently considered a cardinal and prominent symptom of COVID-19 [54,55], and may be the only presenting symptom [56] and is commonly associated with loss of taste (dysgeusia) [57]. Anosmia results from damage to the olfactory epithelium or, more commonly, the central olfactory pathway [58]. It has been observed in previous human coronavirus infections; however, the incidence is higher with SARS-CoV-2 infection [59]. Some cross-sectional studies have reported anosmia incidences ranging from 33.9% to 68% [60–63]. Among 114 swab-positive patients, Klopfenstein et al. [60] reported an anosmia incidence rate of 47%, with a mean duration of 8.9 days.

In this study, the incidence of anosmia was 62%, with a mean duration of 15.9 days and 64% of cases are of severe degree while the incidence of loss of taste was 57%, with a mean duration of 13.9 days and 55% of cases are of severe degree.

The higher percentages and longer durations found in our study may be attributed to mutations of SARS-CoV-2, which result in different genotypes and pathogenicity [61]. Other factors may be related to the currently increased awareness of anosmia in patients with COVID-19 compared to early in the outbreak. Another reason may be related to the varying pathogenicity

**Table 5. (Level 2 A): Scoring Post-recovery COVID-19 long term symptoms persistence (Lower concerns groups).** Scoring method: One point given for each item per symptom (one point for days and one point for age group). Interpretation: Score two for any symptom: follow up is recommended for the patient for each symptom got score 2. **(Level 2 B): Scoring Post-recovery COVID-19 long term symptoms degree of severity (Lower concerns groups).** Scoring method: One point given for each item per symptom (one point for degree of severity and one point for age group) with exception for loss of smell and loss of taste. Interpretation: Score two for any symptom: follow up is recommended for the patient for each symptom got score 2.

| Long term symptom | No. of days (≥) ** | Age group (years) |
|---|---|---|
| Fatigue | 7 | 10-39y |
| Weakness | 8 | 10-39y |
| Muscle aches | 8 | Any age group* |
| Joint pains | 7 | <20y |
| Skin Rash | 5 | Any age group* |
| Headache | 7 | Any age group* |
| Mood changes | 10 | <20y or ≥60y |
| Insomnia | 9 | Any age group* |
| Paraesthesia and anesthesia | 7 | Any age group* |
| Hearing problems | 8 | Any age group* |
| Visual disturbances | 7 | Any age group* |
| Dry eyes | 8 | Any age group* |
| Loss of smell | 16 | Any age group* |
| Loss of taste | 14 | Any age group* |
| Cough | 11 | Any age group* |
| Breathlessness and chest tightness | 9 | Any age group* |
| Lack of appetite | 10 | <20y or ≥60y |
| Nausea | 7 | <20y or ≥60y |
| Diarrhea | 6 | Any age group* |
| Abdominal pain | 6 | <20y or ≥60y |
| **Long term symptom** | **Degree of severity** ** | **Age group (years)** |
| Muscle aches | >2 | Any age group* |
| Joint pains | >1 | Any age group* |
| Skin Rash | >1 | Any age group* |
| Headache | >2 | <20y or≥60y |
| Mood changes | >2 | Any age group* |
| Insomnia | >2 | 20-39y |
| Paraesthesia and anesthesia | >1 | Any age group* |
| Hearing problems | >1 | Any age group* |
| Visual disturbances | >1 | Any age group* |
| Dry eyes | >1 | <20y or 40-59y |
| ***Loss of smell | 3 in severity plus 9.7 days**** | <20y |
| ***Loss of taste | 3 in severity plus 12 days**** | ≥40y |
| Cough | >1 | 20-39y or ≥60y |
| Breathlessness and chest tightness | >2 | Any age group* |
| Lack of appetite | >2 | 20-39y |
| Nausea | >1 | Any age group* |
| Diarrhea | >1 | Any age group* |
| Abdominal pain | >1 | Any age group* |

*Cases age between 10–84 years.

**Days here are the average number of days recorded for each symptom.

*Cases age between 10–84 years.

**Degree of severity Mild = 1 Moderate = 2 Severe = 3.

****Degree plus persistence duration got one point.

****Days here are the average number of days for the mentioned age group.

of SARS-CoV-2 among humans; however, this remains speculative and needs more supportive evidence. In fact, the incidence of anosmia was as high as 98% in a study by Moein *et al.*, who performed specific olfactory testing in patients positive for COVID-19, and identified a high percentage of patients (63%) who were unaware of their anosmia [62]. Other sensory effects include hearing and visual disturbances (17% and 19%, respectively), which are rarely investigated in those with acute COVID-19 illness. To the best of our knowledge, this is the first investigation to include patients who recovered from COVID-19.

Respiratory symptoms that persisted after acute illness included cough of mild severity (47%, with a mean duration of 11 days) and dyspnea of moderate severity (43%, duration 9 days). In 2020, a study by Garrigues et al. investigating discharged patients with COVID-19 found that the incidence of dyspnea was 42% and there was no difference between those admitted to the intensive care unit and general ward [63]. Gastrointestinal symptoms were significant in our study for lack of appetite (42%), with a mean duration of 9.4 days (moderate severity), while other symptoms, such as nausea (44% [6.9 days]), diarrhea (53%, [6.4 days]), and abdominal pain (50% [6 days]), were all mild.

To our knowledge, this is the first study to investigate post-recovery gastrointestinal symptoms in those with COVID-19. Most centers and hospitals considered COVID-19 case recovered by subsidence of acute symptoms. Follow up of hundreds of thousands of recovered cases is not practical and will add much extra load on health services which are barely coping with acute cases. Furthermore, we aren't able to anticipate which patients will suffer long-term symptoms following their recovery, and if so, does all long-term symptoms cases need follow up. As our study highlights the importance of follow-up of patients who recover from acute illness to identify those who may be more likely to experience long-term symptoms that may require further care and investigation. To solve this issue, we proposed a scoring system for COVID-19 post recovery long-term persistent symptoms according to the degree of severity and the duration of persistence, which is best known to our knowledge, is the first time all over the world to classify and score COVID-19 post recovery long term symptoms (Tables 4 and 5).

## Limitations

This study has few limitations including lack of knowledge of the severity of initial illness, details regarding hospitalization, obstacles to meet all survey participants, and the lack of a control group. In addition, the questionnaire used in this study has four limitations: (1) The first obvious one was the respondent's previous experience filling out questionnaires may have an impact on the overall outcome, (2) who completed the questionnaires? The respondent or the surveyor, (3) to receive truthful responses from respondents who fill out the questionnaires, (4) the ways of online administration of the questionnaire such as Google and Twitter forums. In comparison to an interview room, an on-site survey may generate distraction owing to noise and task.

## Conclusions

This study confirmed that the long-term persistent symptoms are evident among individuals who recover from COVID-19. This should raise awareness of the importance of post-recovery follow-up of cases to manage persistent symptoms and reduce the burden on patients and the community. Further studies, however, are needed to investigate the pathogenesis of these persistent symptoms. As we recorded long term persistent symptoms with different degrees of severity and variable persistence, we propose to classify COVID-19 illness into two phases: acute and chronic so we can consider long term persistent symptoms after recovery from acute illness and not to miss any case. Our proposed scoring system will encourage wider scale

studies to confirm and refine the findings by considering geographical distribution and a larger number of COVID-19 cases. This will help to identify priorities in follow-up among patients according to our proposed scoring system and to avoid prolonged suffering in those who considered recovered from COVID-19 "acute illness".

## Supporting information

**S1 Dataset.**
(XLSX)

**S1 File. The English and Arabic versions of the questionnaire used in the study.**
(DOCX)

## Acknowledgments

The authors thank all health services personnel and volunteers who are on the front lines of the pandemic and have expended great effort to fight the disease all over the world. Special thanks to those who sacrificed their lives to save thousands of others. The authors also thank those who devoted their time and exerted extraordinary effort to create vaccines to ease—if not eliminate—suffering around the world.

## Author Contributions

**Conceptualization:** Mostafa M. Khodeir, Hassan A. Shabana.

**Data curation:** Mostafa M. Khodeir, Hassan A. Shabana, Zafar Rasheed, Abdullah S. Alkhamiss, Mohamed Khodeir, Mohammad S. Alkhowailed, Mansour Alsoghair, Suliman A. Alsagaby, Waleed Al Abdulmonem.

**Formal analysis:** Mostafa M. Khodeir, Hassan A. Shabana, Zafar Rasheed, Abdullah S. Alkhamiss, Mohamed Khodeir, Mohammad S. Alkhowailed, Mansour Alsoghair, Suliman A. Alsagaby, Waleed Al Abdulmonem.

**Funding acquisition:** Mostafa M. Khodeir, Hassan A. Shabana, Waleed Al Abdulmonem.

**Investigation:** Mostafa M. Khodeir, Hassan A. Shabana, Zafar Rasheed, Sami Alharbi, Mansour Alsoghair, Waleed Al Abdulmonem.

**Methodology:** Mostafa M. Khodeir, Hassan A. Shabana, Mohammad S. Alkhowailed, Sami Alharbi, Mansour Alsoghair, Waleed Al Abdulmonem.

**Project administration:** Mostafa M. Khodeir, Hassan A. Shabana, Sami Alharbi.

**Resources:** Mostafa M. Khodeir, Hassan A. Shabana.

**Software:** Mostafa M. Khodeir, Hassan A. Shabana, Mansour Alsoghair.

**Supervision:** Mostafa M. Khodeir, Hassan A. Shabana.

**Validation:** Mostafa M. Khodeir, Hassan A. Shabana, Suliman A. Alsagaby.

**Visualization:** Mostafa M. Khodeir, Hassan A. Shabana, Suliman A. Alsagaby.

**Writing – original draft:** Mostafa M. Khodeir, Hassan A. Shabana, Zafar Rasheed.

**Writing – review & editing:** Mostafa M. Khodeir, Hassan A. Shabana, Zafar Rasheed.

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
