## [Decision Letter · Decision Letter 0]

31 Mar 2021

PONE-D-21-02645

COVID-19: Post recovery long-term symptoms among patients in Saudi Arabia

PLOS ONE

Dear Dr. Rasheed,

Thank you for submitting your manuscript to PLOS ONE. After careful consideration, we feel that it has merit but does not fully meet PLOS ONE’s publication criteria as it currently stands. Therefore, we invite you to submit a revised version of the manuscript that addresses the points raised during the review process.

We look forward to receiving your revised manuscript.

Kind regards,

Tauqeer Hussain Mallhi, Ph.D

Academic Editor

PLOS ONE

Journal Requirements:

3. Please include additional information regarding the survey or questionnaire used in the study and ensure that you have provided sufficient details that others could replicate the analyses. For instance, if you developed a questionnaire as part of this study and it is not under a copyright more restrictive than CC-BY, please include a copy, in both the original language and English, as Supporting Information. Moreover, please include more details on how the questionnaire was pre-tested, and whether it was validated.

"The funders had no role in study design, data collection and analysis, decision to

publish, or preparation of the manuscript."

5. Please include your tables as part of your main manuscript and remove the individual files. Please note that supplementary tables (should remain/ be uploaded) as separate "supporting information" files

Additional Editor Comments (if provided):

Dear Authors, thank you for submitting in Plos One. Your manuscript has been assessed by relevant experts from the field. They found manuscript interesting but raised some concerns in methodology (study instrument validation and translation, sampling technique and methods of recruitment) and interpretation of results. It is requested to please consider the comments of reviewers.

Reviewers' comments:

Reviewer's Responses to Questions

**Comments to the Author**

1. Is the manuscript technically sound, and do the data support the conclusions?

Reviewer #1: Partly

Reviewer #2: Partly

2. Has the statistical analysis been performed appropriately and rigorously? 

Reviewer #1: No

Reviewer #2: No

3. Have the authors made all data underlying the findings in their manuscript fully available?

Reviewer #1: Yes

Reviewer #2: Yes

4. Is the manuscript presented in an intelligible fashion and written in standard English?

Reviewer #1: No

Reviewer #2: Yes

5. Review Comments to the Author

Reviewer #1: In their paper COVID-19: Post recovery long-term symptoms among patients in Saudi Arabia, Khodeir et al aimed to identify long-term symptoms in different body

systems symptoms following COVID-19, their severity, and their duration as a first step in building a system to classify post-recovery long-term symptoms of coronavirus disease 2019 (COVID-19). They conclude that Long-term symptoms after recovery from COVID-19 warrant patient follow-up. The authors propose a classification system as a starting point to guide the identification and follow-up of long-term symptoms post-recovery, and recommend larger-scale studies to broaden the definition of recovery from COVID-19, which appears to have two phases, acute and chronic.

While the goal of this paper is interesting, several improvements can be made to this paper:

1. The methods sections requires expansion, a copy of the questionnaire, its translation etc needs to be placed in the methods.

2. The statistics portion requires more sophisticated analysis. For example, usually the mean is reported with SD, and the median is reported with the interquartile range and/or range. Here the authors report the mean SD and range. Also, to develop a score or classification system, there has to be an end point. For example, what is the goal of this classification? Is there a stage? Is there a score? Is one type better than the others. Several adjustments of baseline characteristics are needed. Saudi versus non-Saudi is not needed, Race, ethnicity, work, etc. may be better. LOTS of improvement may be made to this section

3. The results section needs more details. I think reporting the “classification system” reposting a “scale for it” briefly reporting the populations. Again, improvements need to be made

4. The tables could become one table

5. Figures would help deliver the message better

6. The conclusion is too long, and should not have references or references to tables, this should move to the discussion. The conclusion summarizes your findings, not others.

In summary, while the findings and goal of the study is quite good, the structure of this paper could improve.

Reviewer #2: In the manuscript authored by Mostafa Khodeir et al, the authors reported results obtained from an on-line survey about the post recovery long-term symptoms among patients in Saudi Arabia. Despite manuscript is valuable in the attempt here below I reported my comments and suggestions:

-It is not clear in the method section how subjects have been included. Authors stated: "The estimated sample size was 663, which was obtained by statistical calculation from the official Saudi Ministry of Health website, which announced the number of recovered cases with a 99% confidence level and a 5% margin of error". Thus, why authors did not contacted directly those 663 persons? It is correct the 99% confidence level?

-Thus it is not clear if the 979 subjects who performed the on-line survey were all recovered for COVID-19.

-The group analyzed is manily composed from young adults sice only 2.7% were >50 years. Thus results obtained aremailny reppresentative of the post recovery in young adults patients.

-No statistical analysis is present. Only mean and frequency.

-Again, authors should also mentioned that on-line survey can be affected from the subjective perception of the interviewed.

-Elderlies are highly affected with harmful post recovery long-term symptoms, but this group is missing

-In tables included in the manuscript, when authors mention: "mean" and "frequency" please include also to what they are refering for, as days.

6. PLOS authors have the option to publish the peer review history of their article (what does this mean?). If published, this will include your full peer review and any attached files.

Reviewer #1: No

Reviewer #2: No

---

## [Author Response · Author response to Decision Letter 0]

30 May 2021

Department of Medical Biochemistry,

College of Medicine, 

Qassim University,

Buraidah, KSA. 

May 28, 2021

Tauqeer Hussain Mallhi, Ph.D

Academic Editor

PLOS ONE

Dear Professor Mallhi:

Thank you for giving us the opportunity to revise our manuscript # PONE-D-21-02645, entitled “COVID-19: Post recovery long-term symptoms among patients in Saudi Arabia”. The manuscript has been revised in light of the reviewers’ comments for publication in PLOS ONE. The material contained in the manuscript is original and has not been submitted for publication elsewhere. 

We have addressed all the concerns raised by reviewers as detailed under the enclosed point-by-point reply to the reviewers. 

We continue to express our gratitude to the reviewer’s constructive criticisms, which have improved the quality of the manuscript. We hope that the reviewers will find the answers satisfactory.

Sincerely,

Corresponding Author

Zafar Rasheed, MS., PhD., PGDCA. 

ORCID ID: http://orcid.org/0000-0002-8651-4218

Email: zafarrasheed@qumed.edu.sa

Point-by-Point Response to the Reviewers’ Comments 

Manuscript Number: PONE-D-21-02645

Manuscript title: COVID-19: Post recovery long-term symptoms among patients in Saudi Arabia

Journal: PLOS ONE

We are grateful to the reviewers for their thorough reviews and for constructive comments, which enabled us to further improve our manuscript. Concerns raised have been addressed below. 

Please note that all the changes in the revised manuscript are highlighted. 

REVIEWERS REPORTS:

Review Comments to the Author

Author’s Response: 

We highly appreciate to the reviewers and the editors for their thorough reviews. The manuscript has been revised exactly in the same ways as per of your suggestions. Please note that all changes in the revised manuscript are highlighted. 

Reviewer 1: Report

In their paper COVID-19: Post recovery long-term symptoms among patients in Saudi Arabia, Khodeir et al aimed to identify long-term symptoms in different body systems symptoms following COVID-19, their severity, and their duration as a first step in building a system to classify post-recovery long-term symptoms of coronavirus disease 2019 (COVID-19). They conclude that Long-term symptoms after recovery from COVID-19 warrant patient follow-up. The authors propose a classification system as a starting point to guide the identification and follow-up of long-term symptoms post-recovery, and recommend larger-scale studies to broaden the definition of recovery from COVID-19, which appears to have two phases, acute and chronic. While the goal of this paper is interesting, several improvements can be made to this paper.

Author’s Response: 

We highly appreciate for your excellent review, which definitely helped us to further improve the quality of the manuscript. All suggestions have now been incorporated in the revised manuscript.

Specific Comments

Comment # 1: 

The methods section requires expansion, a copy of the questionnaire, its translation etc. needs to be placed in the methods.

Author’s Response:

We highly appreciate you for your thorough review. As suggested, the used questionnaire and its translation have now been included in the methods subsection ‘Instrument development and measures’ on page 5 and 6 and the complete details of used questionnaire with translation have now been added in table 1 of the revised manuscript on page 7 and 8. 

Comment # 2. 

The statistics portion requires more sophisticated analysis. For example, usually the mean is reported with SD, and the median is reported with the interquartile range and/or range. Here the authors report the mean SD and range. Also, to develop a score or classification system, there has to be an end point. For example, what is the goal of this classification? Is there a stage? Is there a score? Is one type better than the others. Several adjustments of baseline characteristics are needed. Saudi versus non-Saudi is not needed, Race, ethnicity, work, etc. may be better. LOTS of improvement may be made to this section.

Author’s Response:

We highly appreciate for your excellent review and for pointing out these. As suggested, a detailed statistical analysis have now been performed and included in the methods section, results section and also in the discussion sections. Specifically, we have now been added more detailed scoring system with different levels of priorities for patient follow up and cut off values. Please see the newly statistical data in the revised Table 3, 4, 5 and 6 on page 12,13,19-21, respectively and also see the newly added figure 1 and 2. Furthermore, the data on nationality of the studied subject have now been omitted as suggested. Please note table numbers in the revised manuscript are changed. 

Comment # 3. 

The results section needs more details. I think reporting the “classification system” reposting a “scale for it” briefly reporting the populations. Again, improvements need to be made

Author’s Response:

Thanks for your useful suggestion which definitely helped us to further improve the quality of the manuscript. As suggested, more details in the results section have now been included in the revised manuscript. Please see the highlighted text in Tables 2-4 on page 11-13 and also see the added data in figures 1-3. Furthermore, please also see the highlighted text in the results section on page 17, line 3-23 and page 18, line 1-4. 

Comment # 4.

The tables could become one table

Author’s Response:

Thanks. All possible modifications in the tabular presentation have been made in the revised manuscript. In addition, the data have also now been presented in the forms of figures. Please note table numbers in the revised manuscript are changed. 

Comment # 5. 

Figures would help deliver the message better

Author’s Response:

Thanks. As suggested, figures have now been included in the revised manuscript. Please see the data in newly data figures 1-3. 

Comment # 6. 

The conclusion is too long, and should not have references or references to tables, this should move to the discussion. The conclusion summarizes your findings, not others.

Author’s Response:

As suggested, the conclusion section has now been revised and the word count has now been significantly reduced. All text with citations has been shifted and discussed in the discussion section. Please see the highlighted text on page 17, line 3-23, page 18, line 1-4 in the discussion section and also page 21, line 10-20 in the conclusion section of the revised manuscript. 

Comment # 7. 

In summary, while the findings and goal of the study is quite good, the structure of this paper could improve.

Author’s Response:

We highly appreciate your for your constructive criticisms and positive comments, which definitely have improved the quality of the manuscript. 

Reviewer 2: Report

In the manuscript authored by Mostafa Khodeir et al, the authors reported results obtained from an on-line survey about the post recovery long-term symptoms among patients in Saudi Arabia. Despite manuscript is valuable in the attempt here below I reported my comments and suggestions.

Author’s Response:

We are grateful to you for your thorough review and for your constructive comments. All concerns raised have been addressed below. Please note that all major changes in the revised manuscript are highlighted. 

Comment # 1:

It is not clear in the method section how subjects have been included. Authors stated: "The estimated sample size was 663, which was obtained by statistical calculation from the official Saudi Ministry of Health website, which announced the number of recovered cases with a 99% confidence level and a 5% margin of error". Thus, why authors did not contacted directly those 663 persons? It is correct the 99% confidence level?

Author’s Response:

Thanks for your thorough review and pointing out these. As suggested, more details in the methods section have now been provided to clarify how the participants were included. Now we have changed the confidence level to 95% and the details of recalculation of the sample size have been provided on page 9, line 1-4 in the revised manuscript. We are also pleased to inform that our survey response exceeded the calculated sample size in both situations of CI 95% or 99%. We would also like to inform that we had not contacted the cases directly in person because of some obstacles such as (a) social and precautionary measures and restrictions in the country, (b) fear of recovered persons to contact hospital or clinics for reinfection (still deficient information about the disease), (c) difficulties in transportation and feasibility time of the respondents, (d) there is still no clear concept about the need of recovered COVID-19 cases for follow up, therefore we considered the piloted sample (40 cases) to continue the study by online survey not to bother the respondents and also the urgency to know more about the disease to recommend all required measures to manage COVID-19 especially those missed and considered recovered by just recovery from acute symptoms. Now we have clearly mentioned that the survey was conducted online and the limitations of the study have also been clearly defined in the revised methods section on page 6, line 15-22. 

Comment # 2:

Thus it is not clear if the 979 subjects who performed the on-line survey were all recovered for COVID-19. 

Author’s Response:

Yes. All were recovered from acute symptoms of COVID-19. We have now added this information in the results section on page 9, line 17 and 18 of the revised manuscript. 

Comment # 3:

The group analyzed is mainly composed from young adults since only 2.7% were >50 years. Thus results obtained are mainly representative of the post recovery in young adults patients.

Author’s Response:

Yes, this information is 100% correct. This could be explained as the survey being done online which is easier for young. This important information has now been added in the limitations of the study of the revised manuscript. 

Comment # 4:

No statistical analysis is present. Only mean and frequency.

Authors Response:

As suggested, more statistical analysis details have now been included throughout the manuscript including tables and figures. And the details of the used statistical methods have been summarized in the revised manuscript on page 9, line 6-14. 

Comment # 5:

Again, authors should also mentioned that on-line survey can be affected from the subjective perception of the interviewed.

Author’s Response:

We highly appreciate for your thorough review and pointing out this. As suggested, this detail has now been included in the limitations of the study on page 16&17, and also in the limitation of questionnaire on page 6 of the revised manuscript. 

Comment # 6:

Elderlies are highly affected with harmful post recovery long-term symptoms, but this group is missing.

Author’s Response:

Thanks for your suggestion and pointing out this. As suggested, the detail of this missing age group has now been included. Now we performed analysis on the basis of age group distribution where elderlies were characterized as 60 or above years of age. These details are summarized in Table 2 of the revised manuscript on page 11. 

Comment # 7:

In tables included in the manuscript, when authors mention: "mean" and "frequency" please include also to what they are refering for, as days.

Author’s Response:

Thanks. As suggested, the tables have now been revised and the suggested details of referring of days have now been included in the revised tables 3 and 5 on page 12&13 and their related text has now been added and highlighted in the results section on page 9 and 10 and discussed in the discussion section on page 12-13 in the revised manuscript. 

Reviewer's Responses to the Editorial Questions: Author’s Response

Comment # 1: Is the manuscript technically sound, and do the data support the conclusions?

Reviewer #1: Partly

Reviewer #2: Partly

Author’s Response:

We highly appreciate both of the reviewers for their thorough review and constructive criticisms, which helped us to further improved the quality of the manuscript. We have addressed all the concerns raised above and hope that the both of you find the answers satisfactory.

Comment # 2: Has the statistical analysis been performed appropriately and rigorously?

Reviewer #1: No

Reviewer #2: No

Author’s Response:

Thanks. This part has also been significantly improved by addition of more statistical analyzed data. Please see the highlighted text in the methods section, results section and particularly tables and figures of the revised manuscript.

Comment # 3: Have the authors made all data underlying the findings in their manuscript fully available?

Reviewer #1: Yes

Reviewer #2: Yes

Author’s Response:

We highly appreciate for the positive comments. 

Comment # 4: Is the manuscript presented in an intelligible fashion and written in standard English?

Reviewer #1: No

Reviewer #2: Yes

Author’s Response:

Thanks. As suggested by the editorial members, the manuscript has been edited by an expert English Editing Company (Editage) and also has now been revised by one of our English language experts. Now we hope that the revised manuscript will now meet the high standards of the journal and suitable for publication in PLOS ONE. 

EDITORIAL COMMENTS

Comment # 1:

and

Author’s Response:

We highly appreciate the editorial board of PLOS ONE for giving us an opportunity to revise our manuscript. The manuscript has now been revised exactly in the same format as instructed. 

Comment # 2: 

Please provide additional details regarding participant consent. In the ethics statement in the Methods and online submission information, please ensure that you have specified what type you obtained (for instance, written or verbal, and if verbal, how it was documented and witnessed). If your study included minors, state whether you obtained consent from parents or guardians. If the need for consent was waived by the ethics committee, please include this information.

Author’s Response:

Thanks. Informed consent statement has now been well defined and included in the methods section on page 4, line 24; page 5, lines 1 and line 5-8 in the revised manuscript. Please note all changes made in the revised manuscript are highlighted. 

Comment # 3:

Please include additional information regarding the survey or questionnaire used in the study and ensure that you have provided sufficient details that others could replicate the analyses. For instance, if you developed a questionnaire as part of this study and it is not under a copyright more restrictive than CC-BY, please include a copy, in both the original language and English, as Supporting Information. Moreover, please include more details on how the questionnaire was pre-tested, and whether it was validated.

Author’s Response:

As suggested by you and one of the reviewers, the complete details of the questionnaire have now been included and its copy with Arabic translation has been inserted in the methods section on page 7 and 8. Furthermore, all possible details of the questionnaire including its validation have now been added in the methods subsection ‘Instrument development and measures’ on page 5 and 6 of the revised manuscript. 

Comment # 4: 

Thank you for stating the following financial disclosure: "The funders had no role in study design, data collection and analysis, decision to publish, or preparation of the manuscript." At this time, please address the following queries:

Comment # 4 (a): Please clarify the sources of funding (financial or material support) for your study. List the grants or organizations that supported your study, including funding received from your institution. 

Author’s Response:

We as authors hereby declared that this study has no association with any of the funding agencies. 

Comment # 4 (b): State what role the funders took in the study. If the funders had no role in your study, please state: “The funders had no role in study design, data collection and analysis, decision to publish, or preparation of the manuscript.” 

Author’s Response:

We declared that “The funders had no role in study design, data collection and analysis, decision to publish, or preparation of the manuscript”. This statement has also now been mentioned in the ‘Funding support’ section of the revised manuscript on page 22, line 14-16. 

Comment # 4 (c): If any authors received a salary from any of your funders, please state which authors and which funders. 

Author’s Response:

None.

Comment # 4 (d): If you did not receive any funding for this study, please state: “The authors received no specific funding for this work.” Please include your amended statements within your cover letter; we will change the online submission form on your behalf

Authors Response:

Thanks. We hereby declared that “The authors received no specific funding for this work.”

Comment # 5:

Please include your tables as part of your main manuscript and remove the individual files. Please note that supplementary tables (should remain/ be uploaded) as separate "supporting information" files

Authors Response:

Thanks. As suggested, all tables have now been inserted in the main manuscript. 

Comment # 6: 

Additional Editor Comments (if provided): 

Dear Authors, thank you for submitting in Plos One. Your manuscript has been assessed by relevant experts from the field. They found manuscript interesting but raised some concerns in methodology (study instrument validation and translation, sampling technique and methods of recruitment) and interpretation of results. It is requested to please consider the comments of reviewers.

Author’s Response:

Thank you for giving us the opportunity to revise this manuscript. The manuscript has been revised in light of the reviewers’ and editorial comments for publication and we believe that the revised manuscript will now meet the high standards of the journal and suitable for publication in “PLOS ONE”. 

Corresponding Author

Zafar Rasheed, MS, PhD, PGDCA.

ORCID ID: http://orcid.org/0000-0002-8651-4218

---

## [Decision Letter · Decision Letter 1]

28 Jul 2021

PONE-D-21-02645R1

COVID-19: Post recovery long-term symptoms among patients in Saudi Arabia

PLOS ONE

Dear Dr. Rasheed,

Thank you for submitting your manuscript to PLOS ONE. After careful consideration, we feel that it has merit but does not fully meet PLOS ONE’s publication criteria as it currently stands. Therefore, we invite you to submit a revised version of the manuscript that addresses the points raised during the review process.

We look forward to receiving your revised manuscript.

Kind regards,

Tauqeer Hussain Mallhi, Ph.D

Academic Editor

PLOS ONE

Journal Requirements:

Additional Editor Comments (if provided):

Dear Authors, thank you for revising the manuscript. Your manuscript has been again assessed by relevant experts. They found manuscript interesting but raised few more concerns in methodology (pilot study, sampling methods etc) and interpretation of results. It is requested to please consider the comments of reviewers.

Reviewers' comments:

Reviewer's Responses to Questions

**Comments to the Author**

1. If the authors have adequately addressed your comments raised in a previous round of review and you feel that this manuscript is now acceptable for publication, you may indicate that here to bypass the “Comments to the Author” section, enter your conflict of interest statement in the “Confidential to Editor” section, and submit your "Accept" recommendation.

Reviewer #2: All comments have been addressed

Reviewer #3: All comments have been addressed

Reviewer #4: (No Response)

Reviewer #5: (No Response)

2. Is the manuscript technically sound, and do the data support the conclusions?

Reviewer #2: Yes

Reviewer #3: Partly

Reviewer #4: Yes

Reviewer #5: Partly

3. Has the statistical analysis been performed appropriately and rigorously? 

Reviewer #2: Yes

Reviewer #3: Yes

Reviewer #4: Yes

Reviewer #5: Yes

4. Have the authors made all data underlying the findings in their manuscript fully available?

Reviewer #2: Yes

Reviewer #3: Yes

Reviewer #4: Yes

Reviewer #5: No

5. Is the manuscript presented in an intelligible fashion and written in standard English?

Reviewer #2: Yes

Reviewer #3: No

Reviewer #4: Yes

Reviewer #5: Yes

6. Review Comments to the Author

Reviewer #2: Authors provided most of the suggestions and modification requested. Now the manuscript has been improved, thus I suggest it for a pubblication in PLOSone

Reviewer #3: Appreciating the authors for their response to the required comments. However, the manuscript still needs English language editing.

Reviewer #4: COVID-19: Post recovery long-term symptoms among 2 patients in Saudi Arabia

Review

Shorten the paper to around 6000 words.

Questionnaire need not be shown in the manuscript

Rewrite the manuscript with all results in one section.

Reviewer #5: With this ongoing World War against this COVID-19 thing, this manuscript addresses a hot topic in which any effort in this concern is highly appreciated. The study is generally scientifically sound however; some important points need to be clarified. These include (on the revised version of the submission):

1) Page 5, lines 1-2: The authors wrote: "A survey was administered between September and October 2020 using Google Forms and Twitter as a forum".

Well, the conclusions based on the answers extracted from a survey on the social media platform are usually taken with extreme caution. So, this point has to be CLEARLY mentioned in the Limitation(s) Section.

2) Page 6, lines 1-2: The authors wrote "The results of the pilot study were not included in any subsequent research". OK, the result of validity and reliability test of the questionnaire has to be included.

3) Page9, line1: OK, here the readers are definitely going to get confused about how the author managed to get this particular figure (384) as a sample size beside, line 17 on the same page 9 tells a completely different story about the number of subjects recruited (979) with no clue whatsoever. So the authors need to mention the whole story of the sample size calculation.

4) Page 16; Line 23: The authors wrote: “This study has few limitations such as”.

OK here is the thing; I encourage the authors to avoid this kind of open statements which include “such as and/or etc.” So, I do urge the authors to precisely count down all these limitations. Speaking of limitation, I suggest gathering all these limitations under one separate section (subtitle).

5) Tables 4 to 6B: (N) is to be mentioned.

7. PLOS authors have the option to publish the peer review history of their article (what does this mean?). If published, this will include your full peer review and any attached files.

Reviewer #2: No

Reviewer #3: **Yes: **Walid Kamal Abdelbasset

Reviewer #4: **Yes: **Asharaf Abdul Salam

Reviewer #5: No

---

## [Author Response · Author response to Decision Letter 1]

3 Aug 2021

Point-by-Point Re-Response to the Reviewers’ Comments 

Manuscript Number: PONE-D-21-02645R1

Manuscript title: COVID-19: Post recovery long-term symptoms among patients in Saudi Arabia

Journal: PLOS ONE

We are grateful to the reviewers and editors for their thorough reviews and for constructive comments, which enabled us to further re-improve our manuscript. Concerns raised have been re-addressed below. 

Please note that all major changes in the Re-revised manuscript are highlighted. 

REVIEWERS' COMMENTS:

Reviewer's Responses to Questions

Comments to the Author

1. If the authors have adequately addressed your comments raised in a previous round of review and you feel that this manuscript is now acceptable for publication, you may indicate that here to bypass the “Comments to the Author” section, enter your conflict of interest statement in the “Confidential to Editor” section, and submit your "Accept" recommendation.

Reviewer # 2: All comments have been addressed

Author’s Response: We highly appreciate your positive feedback. 

Reviewer # 3: All comments have been addressed

Author’s Response: We highly appreciate your positive feedback.

Reviewer # 4: (No Response)

Author’s Response: We highly appreciate your thorough review, which helped us to further improve the quality of the manuscript. 

Reviewer # 5: (No Response)

Author’s Response: We highly appreciate your thorough review, which helped us to further improve the quality of the manuscript.

2. Is the manuscript technically sound, and do the data support the conclusions?

Reviewer # 2: Yes

Author’s Response: We highly appreciate your positive feedback. 

Reviewer # 3: Partly

Author’s Response: We highly appreciate for the positive response. The manuscript has now been re-revised in light of your comments. 

Reviewer # 4: Yes

Author’s Response: We highly appreciate your positive feedback. 

Reviewer # 5: Partly

Author’s Response: We highly appreciate your positive feedback. The manuscript has now been re-revised in light of your comments. 

3. Has the statistical analysis been performed appropriately and rigorously?

Reviewer # 2: Yes

Author’s Response: We highly appreciate your positive feedback.

Reviewer # 3: Yes

Author’s Response: We highly appreciate your positive feedback. 

Reviewer # 4: Yes

Author’s Response: We highly appreciate your positive response. 

Reviewer # 5: Yes

Author’s Response: We highly appreciate your positive response. 

4. Have the authors made all data underlying the findings in their manuscript fully available?

Reviewer # 2: Yes

Author’s Response: We highly appreciate your positive feedback. 

Reviewer # 3: Yes

Author’s Response: We highly appreciate your positive feedback. 

Reviewer # 4: Yes

Author’s Response: We highly appreciate your positive feedback. 

Reviewer # 5: No

Author’s Response: We highly appreciate your thorough reviews. The manuscript has now been re-revised as per of your suggestions. 

5. Is the manuscript presented in an intelligible fashion and written in standard English?

Reviewer # 2: Yes

Author’s Response: We highly appreciate your positive feedback.

Reviewer # 3: No

Author’s Response: We highly appreciate your thorough reviews. The manuscript has now been re-revised as per of your suggestions.

Reviewer # 4: Yes

Author’s Response: We highly appreciate your positive feedback.

Reviewer # 5: Yes

Author’s Response: We highly appreciate your positive feedback.

6. Review Comments to the Author

Reviewer # 2: Authors provided most of the suggestions and modification requested. Now the manuscript has been improved, thus I suggest it for a publication in PLOSone

Author’s Response: Many thanks for your positive response. 

Reviewer # 3: Appreciating the authors for their response to the required comments. However, the manuscript still needs English language editing.

Author’s Response: We highly appreciate your positive feedback. This manuscript has been edited by an expert English Editing Company Editage and also has now been Re-revised by one of our English language experts. 

Reviewer # 4: COVID-19: Post recovery long-term symptoms among 2 patients in Saudi Arabia.

Review report: 

(a) Shorten the paper to around 6000 words.

Author’s Response: As suggested, the manuscript has now been reduced to ~5500 words. 

(b) Questionnaire need not be shown in the manuscript.

Author’s Response: As suggested, the questionnaire has now been omitted from the re-revised manuscript. Respectfully, we would like to inform that we have placed the questionnaire in the previous version of this manuscript as per instructions of reviewer # 1, which has now been presented as a supplementary file. 

(c) Rewrite the manuscript with all results in one section.

Author’s Response: Thanks. All results have now been presented in one section. 

Reviewer # 5: With this ongoing World War against this COVID-19 thing, this manuscript addresses a hot topic in which any effort in this concern is highly appreciated. The study is generally scientifically sound however; some important points need to be clarified. These include (on the revised version of the submission):

1) Page 5, lines 1-2: The authors wrote: "A survey was administered between September and October 2020 using Google Forms and Twitter as a forum". Well, the conclusions based on the answers extracted from a survey on the social media platform are usually taken with extreme caution. So, this point has to be CLEARLY mentioned in the Limitation(s) Section.

Author’s Response: We highly appreciate your positive feedback. As suggested, the limitation section has now been modified by taking caution. The ways of questionnaire administration have now been well discussed in the limitations of the study on page 18, line 12-21 in the re-revised manuscript.

2) Page 6, lines 1-2: The authors wrote "The results of the pilot study were not included in any subsequent research". OK, the result of validity and reliability test of the questionnaire has to be included.

Author’s Response: Excellent suggestion. As suggested, the validity and reliability test of the questionnaire have now been included in the revised manuscript on page 6, line 1-9 and their associated references have also been cited. Please see the newly added references # 36 and 37 in the reference list on page 24. 

3) Page 9, line 1: OK, here the readers are definitely going to get confused about how the author managed to get this particular figure (384) as a sample size beside, line 17 on the same page 9 tells a completely different story about the number of subjects recruited (979) with no clue whatsoever. So the authors need to mention the whole story of the sample size calculation.

Author’s Response: Thanks for your excellent suggestion. As suggested, this useful information of sample size has now been added in the revised manuscript on page 7, line 4-9. 

4) Page 16; Line 23: The authors wrote: “This study has few limitations such as”.

OK here is the thing; I encourage the authors to avoid this kind of open statements which include “such as and/or etc.” So, I do urge the authors to precisely count down all these limitations. Speaking of limitation, I suggest gathering all these limitations under one separate section (subtitle).

Author’s Response: Thanks. As suggested, all limitations have now been gathered and discussed in the last part of the discussion section on page 18, line 12-21 in the revised manuscript. 

5) Tables 4 to 6B: (N) is to be mentioned.

Author’s Response: Thanks. All tables and figures have now been cited in the results section of the re-revised manuscript. 

7. PLOS authors have the option to publish the peer review history of their article (what does this mean?). If published, this will include your full peer review and any attached files.

Do you want your identity to be public for this peer review? For information about this choice, including consent withdrawal, please see our Privacy Policy.

Reviewer #2: No

Reviewer #3: Yes: Walid Kamal Abdelbasset

Reviewer #4: Yes: Asharaf Abdul Salam

Reviewer #5: No

Author’s Response: We highly appreciate all the reviewers for their excellent review, which have improved the quality of the manuscript.

We believe that the Re-revised manuscript will now meet the high standards of the journal and suitable for publication in “PLOS ONE”. 

Corresponding Author

Zafar Rasheed, MS, PhD, PGDCA.

ORCID ID: http://orcid.org/0000-0002-8651-4218

---

## [Decision Letter · Decision Letter 2]

13 Sep 2021

PONE-D-21-02645R2COVID-19: Post recovery long-term symptoms among patients in Saudi ArabiaPLOS ONE

Dear Dr. Rasheed,

Thank you for submitting your manuscript to PLOS ONE. After careful consideration, we feel that it has merit but does not fully meet PLOS ONE’s publication criteria as it currently stands. Therefore, we invite you to submit a revised version of the manuscript that addresses the points raised during the review process.

We look forward to receiving your revised manuscript.

Kind regards,

Tauqeer Hussain Mallhi, Ph.D

Academic Editor

PLOS ONE

Journal Requirements:

Additional Editor Comments (if provided):

Dear Authors, thank you for revising the draft. This manuscript has improved substantially. However, referee raised few concerns over the limitation of the project. Please address the comments and submit the draft at your earliest.

Reviewers' comments:

Reviewer's Responses to Questions

**Comments to the Author**

1. If the authors have adequately addressed your comments raised in a previous round of review and you feel that this manuscript is now acceptable for publication, you may indicate that here to bypass the “Comments to the Author” section, enter your conflict of interest statement in the “Confidential to Editor” section, and submit your "Accept" recommendation.

Reviewer #2: All comments have been addressed

Reviewer #3: All comments have been addressed

Reviewer #4: All comments have been addressed

Reviewer #5: (No Response)

2. Is the manuscript technically sound, and do the data support the conclusions?

Reviewer #2: Yes

Reviewer #3: Yes

Reviewer #4: Yes

Reviewer #5: Partly

3. Has the statistical analysis been performed appropriately and rigorously? 

Reviewer #2: Yes

Reviewer #3: Yes

Reviewer #4: Yes

Reviewer #5: N/A

4. Have the authors made all data underlying the findings in their manuscript fully available?

Reviewer #2: Yes

Reviewer #3: Yes

Reviewer #4: No

Reviewer #5: No

5. Is the manuscript presented in an intelligible fashion and written in standard English?

Reviewer #2: Yes

Reviewer #3: Yes

Reviewer #4: Yes

Reviewer #5: Yes

6. Review Comments to the Author

Reviewer #2: (No Response)

Reviewer #3: All required comments have been addressed. The manuscript is presented in an intelligible fashion and written in standard English?I have no further comments. Congrats.

Reviewer #4: COVID-19: Post recovery long-term symptoms among patients in Saudi Arabia

1. A total of 979 patients 10 recovered from COVID-19 in Saudi Arabia in the study period, of whom 53% were male and 47% were female.

Sample size is appreciated as such studies from Saudi Arabia are rare, but not this male-female proportions. It is stated that males are affected more than females. Here the difference is not large.

3–5 min (P5; L8) may change to 3-5 minutes

Instrument development and measures section is one paragraph. Consider splitting into two or even three.

Final questionnaire was structured into 6 sections 10 addressing specific symptoms (P6 L 9-10) – In case the authors adopted any classification system, the same may be referred.

In P7 the following statements need explanations a. a total of 992 subjects were approached b. 13 subjects were omitted as they were made invalid selection and c. 979 subjects were recruited. Whether used a sample frame or a followed a list of subjects. Sample representativeness needs to be explained.

Of the 979 respondents’ patients (P8)- use of ‘

respectively) (Table 3) (P8; L18)– Avoid two parentheses together

Furthermore, we also found 21 a statistically significant relation of age with the presence of post-recovery COVID-19 long term 22 symptoms degree of severity and/or persistence, such as weakness degree (p= 0.003), persistence (p=0.001), lack of appetite degree (p=0.02), persistence (p=0.003), Insomnia degree (p=0.01), 9 1 loss of smell degree (p=0.002), loss of taste degree and Headache degree (p=0.04 each), cough 2 degree (p=0.01) significant correlation was found between age and persistence of symptoms as 3 fatigue (p=0.004), joint pains (p=0.01), mood changes (p=0.03), nausea (p=0.002) and 4 abdominal pain (p=0.02)( Table 2&3) – Please check the tables and clear confusions by including age in the table titles and columns.

The proposed scoring system in this study can be delivered online for post discharge follow-up, and the score of each case can be automatically calculated. The case that will reach the proposed score should be invited for follow up in the clinic, keeping in mind high concerns groups (Table 4 level 1A&B) are first priority follow up group, while lower concerns groups (Table 5 level 2 A&B) are of second priority follow up. The cases not reached the proposed score no need for clinic follow up and should be reassured as mostly their symptoms be self-limited. By this scoring system we can easily pickup patients of concern to be followed up and also decrease load on health services as much as possible. Why these lines stand different????

Table 2 shows a wide range days (upto 120 days). It would be nice to explain column by column. You may use either mean days or median days. It is nice to state the statistical test performed for reaching out the p value.

Results need to be further elaborated for readers to understand.

Use smaller paragraphs in discussions. I mean, break ideas into paragraphs.

we can’t expect who will suffer (P17, L 9) may be reworded

Our scoring system will help to broaden the view of the scoring system that used to classify the acute cases of COVID-19 into mild, moderate, severe, or critical which proposed by The Chinese National Health Commission [64]. We aimed by our scoring system to map and score, as an initial step to build a scoring system, the recorded long-term symptoms to avoid missing such cases who may suffer a sequalae later on, keeping in mind the exact pathogenesis is still unclear. Our proposed scoring system can be delivered online for post discharge follow-up, and can be automatically calculated the score of each case. The case that will reach the proposed score (score 2) should invited for follow up in the clinic, keeping in mind high concerns groups are first priority follow up group, while lower concerns groups are of second priority follow up. The cases not reached the proposed score no need for clinic follow up and should be reassured as mostly their symptoms be self-limited. By this scoring system we can easily pickup patients of concern to be followed up and also decrease load on health services as much as possible. Again, this scoring system could be applied online weekly and so we can get a wider scale and broader view on behavior of these long-term symptoms. Our proposed scoring system and categorization of patients into high concern and lower concern groups may considered as an initial step that help and encourage a wider scale studies in different countries to confirm and refine the findings by considering geographical distribution and a larger number of COVID-19 cases. This will help to identify priorities in follow-up among patients according to longer-term symptoms and to avoid prolonged suffering in those who recover from COVID-19. (P17-18). Do not match with the discussions, that is wrongly placed.

Acknowledgments The authors thank all health services personnel and volunteers who are on the front lines of the pandemic and have expended great effort to fight the disease all over the world. Special thanks to those who sacrificed their lives to save thousands of others. The authors also thank those who devoted their time and exerted extraordinary effort to create vaccines to ease—if not eliminate— 22 suffering around the world. Authors may acknowledge those who helped them in this research and manuscript preparation.

Reviewer #5: Because they are too many and significantly affect the reliability of all the conclusions and recommendations drawn, all limitations are to be gathered in a separate section under the subtitle "LIMITATIONS" and NOT as a paragraph in the "Discussion" section.

7. PLOS authors have the option to publish the peer review history of their article (what does this mean?). If published, this will include your full peer review and any attached files.

Reviewer #2: No

Reviewer #3: **Yes: **Walid Kamal Abdelbasset

Reviewer #4: **Yes: **Asharaf Abdul Salam

Reviewer #5: No

---

## [Author Response · Author response to Decision Letter 2]

8 Oct 2021

Point-by-Point Re-Response to the Reviewers’ Comments 

Manuscript Number: PONE-D-21-02645R2

Manuscript title: COVID-19: Post recovery long-term symptoms among patients in Saudi Arabia

Journal: PLOS ONE

We are grateful to all the reviewers for their thorough reviews and for constructive comments, which enabled us to further re-improve our manuscript. Concerns raised have been re-addressed below. 

Please note that all major changes in the Re-revised manuscript are highlighted. 

REVIEWERS' COMMENTS:

Reviewer # 1: Final Report Outcome

Author’s Response: We highly appreciate you for the positive feedback

Reviewer # 2: Final Report Outcome

Author’s Response: We highly appreciate you for the positive feedback

Reviewer # 3: Final Report Outcome

Author’s Response: We highly appreciate you for the positive feedback

Reviewer # 4: Report

COVID-19: Post recovery long-term symptoms among patients in Saudi Arabia

Comment 1:

A total of 979 patients 10 recovered from COVID-19 in Saudi Arabia in the study period, of whom 53% were male and 47% were female. Sample size is appreciated as such studies from Saudi Arabia are rare, but not this male-female proportions. It is stated that males are affected more than females. Here the difference is not large.

Author’s Response: 

It has now been corrected on page 8, line 3 in the re-revised manuscript. 

Comment 2:

3–5 min (P5; L8) may change to 3-5 minutes

Author’s Response: 

As suggested, it has now been changed. 

Comment 3: Instrument development and measures section is one paragraph. Consider splitting into two or even three.

Author’s Response: 

As suggested, this paragraph has now been splitted into two subsections on page 5 and 6 in the re-revised manuscript. 

Comment 4: Final questionnaire was structured into 6 sections 10 addressing specific symptoms (P6 L 9-10) – In case the authors adopted any classification system, the same may be referred.

Author’s Response: 

We mentioned that ‘the questionnaire used in this study was created using a combination of published literature and the literature used was well cited on page 5, line 14 and 15. 

Comment 5: In P7 the following statements need explanations a. a total of 992 subjects were approached b. 13 subjects were omitted as they were made invalid selection and c. 979 subjects were recruited. Whether used a sample frame or a followed a list of subjects. Sample representativeness needs to be explained.

Author’s Response: 

Thanks. Sample representativeness has now been mentioned and highlighted on page 7, line 8 in the re-revised manuscript. All subjects were Saudi citizens. 

Comment 6: Of the 979 respondents’ patients (P8)- use of ‘

Author’s Response: It has now been corrected on page 8, line 2. 

Comment 7: respectively) (Table 3) (P8; L18) – Avoid two parentheses together

Author’s Response: Thanks, this has also now corrected.

Comment 8: Furthermore, we also found 21 a statistically significant relation of age with the presence of post-recovery COVID-19 long term 22 symptoms degree of severity and/or persistence, such as weakness degree (p= 0.003), persistence (p=0.001), lack of appetite degree (p=0.02), persistence (p=0.003), Insomnia degree (p=0.01), 9 1 loss of smell degree (p=0.002), loss of taste degree and Headache degree (p=0.04 each), cough 2 degree (p=0.01) significant correlation was found between age and persistence of symptoms as 3 fatigue (p=0.004), joint pains (p=0.01), mood changes (p=0.03), nausea (p=0.002) and 4 abdominal pain (p=0.02)( Table 2&3) – Please check the tables and clear confusions by including age in the table titles and columns.

Author’s Response: 

We have now re-checked both of these tables and the mean (±SD) age of the studied subjects in years has now been added in their titles in the re-revised manuscript as per of your suggestions. 

Comment 9: The proposed scoring system in this study can be delivered online for post discharge follow-up, and the score of each case can be automatically calculated. The case that will reach the proposed score should be invited for follow up in the clinic, keeping in mind high concerns groups (Table 4 level 1A&B) are first priority follow up group, while lower concerns groups (Table 5 level 2 A&B) are of second priority follow up. The cases not reached the proposed score no need for clinic follow up and should be reassured as mostly their symptoms be self-limited. By this scoring system we can easily pickup patients of concern to be followed up and also decrease load on health services as much as possible. Why these lines stand different????

Author’s Response: 

This paragraph is linked to our findings. For more clarity to the readers to understand, we have made some modifications in this paragraph on page 7, line 5-15 in the re-revised manuscript. 

Comment 10: Table 2 shows a wide range days (upto 120 days). It would be nice to explain column by column. You may use either mean days or median days. It is nice to state the statistical test performed for reaching out the p value. Results need to be further elaborated for readers to understand. Use smaller paragraphs in discussions. I mean, break ideas into paragraphs.

Author’s Response:

Thanks. The data in mean days as well as median days have already mentioned in the Table 2. As suggested, the statistical test used to measure p values has now been added in the table legend. Moreover, the discussion sections has now been divided into smaller paragraphs. Please note only major changes in the re-revised manuscript are highlighted. 

Comment 11: we can’t expect who will suffer (P17, L 9) may be reworded. Our scoring system will help to broaden the view of the scoring system that used to classify the acute cases of COVID-19 into mild, moderate, severe, or critical which proposed by The Chinese National Health Commission [64]. We aimed by our scoring system to map and score, as an initial step to build a scoring system, the recorded long-term symptoms to avoid missing such cases who may suffer a sequalae later on, keeping in mind the exact pathogenesis is still unclear. Our proposed scoring system can be delivered online for post discharge follow-up, and can be automatically calculated the score of each case. The case that will reach the proposed score (score 2) should invited for follow up in the clinic, keeping in mind high concerns groups are first priority follow up group, while lower concerns groups are of second priority follow up. The cases not reached the proposed score no need for clinic follow up and should be reassured as mostly their symptoms be self-limited. By this scoring system we can easily pickup patients of concern to be followed up and also decrease load on health services as much as possible. Again, this scoring system could be applied online weekly and so we can get a wider scale and broader view on behavior of these long-term symptoms. Our proposed scoring system and categorization of patients into high concern and lower concern groups may considered as an initial step that help and encourage a wider scale studies in different countries to confirm and refine the findings by considering geographical distribution and a larger number of COVID-19 cases. This will help to identify priorities in follow-up among patients according to longer-term symptoms and to avoid prolonged suffering in those who recover from COVID-19. (P17-18). Do not match with the discussions, that is wrongly placed.

Author’s Response: 

Thanks. This paragraph has now been omitted from the discussion section of the re-revised manuscript. 

Comment 12: Acknowledgments: The authors thank all health services personnel and volunteers who are on the front lines of the pandemic and have expended great effort to fight the disease all over the world. Special thanks to those who sacrificed their lives to save thousands of others. The authors also thank those who devoted their time and exerted extraordinary effort to create vaccines to ease—if not eliminate— 22 suffering around the world. Authors may acknowledge those who helped them in this research and manuscript preparation.

Author’s Response:

The paragraph you copied from the acknowledgement section and pasted above has already mentioned the acknowledgments of all the services, personnel and volunteers that we utilized in the preparation of this manuscript. Respectfully, we are very pleased to inform that all authors themselves participated in this research and in the preparation of manuscript. 

Reviewer # 5: Report 

Because they are too many and significantly affect the reliability of all the conclusions and recommendations drawn, all limitations are to be gathered in a separate section under the subtitle "LIMITATIONS" and NOT as a paragraph in the "Discussion" section.

Author’s Response:

Thanks for your review and for the suggestions. As suggested, study limitations have now been presented in a separate section. 

We believe that this revised manuscript will now meet the high standards of the journal and suitable for publication in “PLOS ONE”. 

Corresponding Author

Zafar Rasheed, MS, PhD, PGDCA.

ORCID ID: http://orcid.org/0000-0002-8651-4218

---

## [Decision Letter · Decision Letter 3]

8 Nov 2021

COVID-19: Post recovery long-term symptoms among patients in Saudi Arabia

PONE-D-21-02645R3

Dear Dr. Rasheed,

We’re pleased to inform you that your manuscript has been judged scientifically suitable for publication and will be formally accepted for publication once it meets all outstanding technical requirements.

Kind regards,

Tauqeer Hussain Mallhi, Ph.D

Academic Editor

PLOS ONE

Additional Editor Comments (optional):

Reviewers' comments:

Reviewer's Responses to Questions

**Comments to the Author**

1. If the authors have adequately addressed your comments raised in a previous round of review and you feel that this manuscript is now acceptable for publication, you may indicate that here to bypass the “Comments to the Author” section, enter your conflict of interest statement in the “Confidential to Editor” section, and submit your "Accept" recommendation.

Reviewer #2: All comments have been addressed

Reviewer #3: All comments have been addressed

Reviewer #5: All comments have been addressed

2. Is the manuscript technically sound, and do the data support the conclusions?

Reviewer #2: Yes

Reviewer #3: Yes

Reviewer #5: Yes

3. Has the statistical analysis been performed appropriately and rigorously? 

Reviewer #2: Yes

Reviewer #3: Yes

Reviewer #5: (No Response)

4. Have the authors made all data underlying the findings in their manuscript fully available?

Reviewer #2: Yes

Reviewer #3: Yes

Reviewer #5: Yes

5. Is the manuscript presented in an intelligible fashion and written in standard English?

Reviewer #2: Yes

Reviewer #3: Yes

Reviewer #5: Yes

6. Review Comments to the Author

Reviewer #2: Authors provided most of the corrections requested from the reviewers and now the manuscript has been highly improved.

Reviewer #3: Appreciating the authors for their complete responses to all comments required by the reviewers. I have no additional comments.

Reviewer #5: (No Response)

7. PLOS authors have the option to publish the peer review history of their article (what does this mean?). If published, this will include your full peer review and any attached files.

Reviewer #2: No

Reviewer #3: **Yes: **Walid Kamal Abdelbasset

Reviewer #5: No

---

## [Editor Report · Acceptance letter]

15 Nov 2021

PONE-D-21-02645R3 

COVID-19: Post recovery long-term symptoms among patients in Saudi Arabia 

Dear Dr. Rasheed:

I'm pleased to inform you that your manuscript has been deemed suitable for publication in PLOS ONE. Congratulations! Your manuscript is now with our production department. 

Kind regards, 

on behalf of

Dr. Tauqeer Hussain Mallhi 

Academic Editor

PLOS ONE